# Perceptions of Sheep Farmers and District Veterinarians towards Sheep Disease Management in New South Wales, Australia

**DOI:** 10.3390/ani14081249

**Published:** 2024-04-22

**Authors:** Jessica Boyd-Weetman, Lauren Alam, Om Dhungyel, Wendy I. Muir

**Affiliations:** 1School of Life and Environmental Science, Faculty of Science, The University of Sydney, Camden, NSW 2570, Australia; jboy4411@uni.sydney.edu.au; 2Sydney School of Veterinary Science, Faculty of Science, The University of Sydney, Camden, NSW 2570, Australia; lala5275@uni.sydney.edu.au (L.A.); om.dhungyel@sydney.edu.au (O.D.)

**Keywords:** sheep farmers, district veterinarians, sheep disease management, biosecurity, sheep health, Australian sheep producers, communicating with farmers

## Abstract

**Simple Summary:**

The perceptions and practices of sheep farmers with regards to their knowledge of sheep health management on their farms can have a significant impact on the overall health, welfare, and production of their animals and on the economics of their farming enterprise. District veterinary services play an important role in providing technical services and support to the farmers. Survey studies were conducted to understand and compare the practices and perceptions of sheep famers and district veterinarians (DVs) in NSW, Australia, to sheep health and its management. The findings illustrated differences between the farmer and DV perception of sheep health and diseases of greatest concern and provided valuable insights into their different observations and practices. For example, sheep producers were most concerned with parasites whereas for DVs sheep husbandry advice was most concerning. However, both groups ranked nutrition advice as the second most concerning area related to sheep health. Of particular interest were the differences in the communication methods that they prefer which, if more closely aligned, may enable more effective communication. The findings of these studies provide some guidance and recommendations for improving the management of the health, welfare, and overall economic value of sheep farming in NSW.

**Abstract:**

The study objectives were to understand the practices and perceptions of sheep farmers and district veterinarians (DVs) towards sheep health management and, the impact of disease at the farm level in addition to the availability, accessibility, and use of veterinary services. Data were collected using question-based surveys, distributed online and in-person to sheep farmers (45 respondents) and DVs (25 respondents). Most farmers were male, ≥51 years old, who placed a high priority on the health and welfare of their sheep. For disease prevention most farmers vaccinated their sheep (91%) and 86.7% had a farm biosecurity plan, although its components and their application varied, e.g., the isolation of new or sick sheep. Fencing costs were most frequently identified (70.5% respondents) as a financial concern for sheep farmers. Their most common sources of information about disease control and prevention were DVs (66.7%), private veterinarians (60.0%), the internet (42.9%), rural suppliers (35.7%), and farmers/neighbours (28.6%). Fifty-eight percent of farmers reported a long distance from veterinarian services. Farmers preferred to receive information via email (77.8%), whereas 56% of DVs preferred to share information via phone call. This discrepancy presents an opportunity to align these mechanisms more closely for effective dissemination of information and increased producer engagement.

## 1. Introduction

Management of sheep health can have a significant impact on animal welfare, human welfare, and economic returns for individual farmers and the sheep industry as a whole [1]. The perspectives of the veterinary and sheep industry sectors about the management of sheep diseases are key to addressing the needs of individual farmers and to improve flock health [2]. As illustrated in the dairy industry, veterinarians are an important source of information for livestock producers by providing advice on animal health [2]. They also aid in dispersing information to encourage the adoption of recommended practices designed to improve the health of production animals, such as sheep and dairy cattle [3]. Therefore, frequent, and effective contact between farmers and veterinarians can be pivotal for improving animal health, welfare, and farm productivity. Despite this there appears to be some constraints with the contact between these two groups. 

A UK study of 2500 sheep producers found that only 67% of sheep farmers contacted their veterinarian when emergency treatment of sick sheep was required and, only 20% had regular contact with their veterinarian [4]. In Australia, veterinary services are highly regarded but underutilized by producers, with low levels of veterinary contact reported when seeking general information and/or reporting disease outbreaks [5,6]. This could be related to accessibility. For example, Western Australian sheep and cattle farmers identified that access to veterinarians was difficult due to the remoteness of farms and the distance between producers and veterinary services [7]. Further, many sheep producers consider their farm to be unique and as such they make management decisions based on their own circumstances, agricultural context, beliefs, and goals [4]. In this context their engagement with veterinarians for advice is tempered by their concerns of inconsistent service, limited expertise of veterinarians with sheep farming and the independence of their advice [4]. Sheep farmers have a general impression that veterinarians lack a broad knowledge of sheep farming [4] and that they may not be comfortable with discussing broader topics including agronomy, nutrition, genetics and economics [8].

Additionally, when veterinarians and farmers communicate, a lack of understanding of the farmer’s perception by the veterinarian can also act as an impediment. For example, there may be a mismatch between what a farmer values in herd health programs and the veterinarian’s perception of their values [9]. Likewise, veterinarians in Quebec were found to misjudge the perceived importance and usefulness of biosecurity by dairy farming clients [3]. Furthermore, effective sharing of knowledge and advice requires professional communication styles that match more closely with those preferred by the farmer or client [9]. Studies exploring the veterinarian–producer relationship in the Australian context are limited. An example of one such study, but from the Australian equine industry, identified that engagement of veterinarians for the management of Hendra virus was linked to the horse owner having more horses, being older, having a ‘duty of care’ for other people working with horses, and deriving their main income from a horse related business [10].

Sheep producers are perceived to be self-sufficient in dealing with disease within their flocks [11]. They are accustomed to dealing with endemic disease with little outside assistance, but this may contribute to a limited understanding of the overall impact of diseases [11]. Likewise, while information on best practice disease management is available to producers, Australian-based studies have highlighted variable on-farm adoption of these practices [12,13,14,15,16,17]. Recent studies exploring the use of vaccines on Australian sheep farms including immunization against clostridial diseases, caseous lymphadenitis (CLA), and Ovine Johne’s disease (OJD) [12,13,17,18] report a dissociation between recommended veterinary advice and farmer compliance. For example, while 43% of Australian sheep producers use vaccination to prevent CLA, only 12% followed the recommended vaccination procedures [18]. Similarly, in the Australian state of New South Wales (NSW) it was reported in 2020 that 96% of sheep producers vaccinated their lambs against clostridial disease but 23% of crossbred and 17% of Merino lamb producers did not vaccinate according to the instructions [13]. 

Modelling by Ritter et al. (2017), explored the common factors that influence the on-farm adoption of recommended management practices [19]. Aside from the farmer’s demographics, such as age, gender, cultural and educational background, adoption is affected by the farmer’s perception of their own ability to make changes, their awareness of the problem, and the feasibility of the proposed changes as dictated by economic and regulatory forces. In the sheep industry, a few studies have ascertained information about specific diseases or aspects of sheep management [13,18], but no studies have explored farmer perceptions on overall sheep health management or assessed veterinary perceptions of the management of diseases on sheep farms. 

It is apparent that there is more to understand about the most effective methods of communication with farmers to improve on-farm management of sheep health. Therefore, a survey of Australian sheep farmers designed to understand their preferences regarding modality and sheep health related topics in which they would like more information was undertaken. Considering the knowledge gaps highlighted above, this study was designed to:(1)Describe sheep producer practices and perceptions towards sheep health management and, their concern about the impact of disease on sheep welfare, mortality, and farm finances.(2)Understand sheep producer access to and use of veterinary services and technical information.(3)Investigate and compare the perceptions of NSW district veterinarians (DVs) and sheep farmers with regards to their knowledge of sheep health and, their perspective of the primary disease issues affecting sheep.(4)Compare the methods used to disseminate information to sheep producers by NSW DVs with the producers’ preferred method of receiving this information.

## 2. Materials and Methods

This study consisted of two survey questionnaires, one designed for sheep farmers in NSW and the second for NSW based DVs. Both survey questionnaires are included in the Appendix A. As the questionnaires were part of a larger study not all of the data collected are included in this report. 

### 2.1. Questionnaire Design 

#### 2.1.1. Sheep Producers

To participate in this survey participants needed to identify as sheep producers with a minimum of 50 sheep, be located within the state of NSW, Australia, and consent to participate in the study. The survey consisted of 76 questions, 40 were close-ended, 24 were open and 12 were semi-closed questions, the latter being where the respondent selected an answer from a list of options and could also provide a written answer by selecting the option of “other”. The questions were designed to collect information about the sheep producer, the property demographics, disease history, and disease management practices, including biosecurity measures, over the 3-year period of 2019–2021. The producer’s perceived accessibility to and use of veterinary services and technical information was also collected. Further questions focused on the producer’s knowledge and perceptions of disease management in addition to the impact of disease on production costs and sheep welfare on their farm. 

#### 2.1.2. District Veterinarians

To participate in the DV survey participants were required to be currently employed as a NSW DV for a Local Land Services (LLS) district and to consent to being involved in the study. District veterinarians were targeted because they are veterinarians employed by the NSW state government to support the farming communities and they service distinct regions of the state. The DV survey consisted of five semi-closed questions. The questions were designed to collect information on the DV perception of areas in which sheep producers required more knowledge, the reasons they were called out to properties, and their method/s of distributing technical information to producers during the 3-year period of 2019–2021.

### 2.2. Ethics Approval 

The surveys were conducted with approval of the Human Research Ethics Committee of The University of Sydney, Approval No. 2021/319 with informed consent of the survey participants. 

### 2.3. Survey Distribution and Data Collection 

#### 2.3.1. Sheep Producers 

The survey of sheep farmers was undertaken between August 2021 and February 2022. The online link for the survey was distributed through a variety of sheep societies in NSW. Additionally, DVs across NSW were contacted via email to assist in distribution of the survey to sheep producers. Six DVs agreed to assist with distribution of the survey, and each contacted up to ten NSW sheep producers. To participate in the survey, respondents needed to identify as sheep producers, be located within the state of NSW, Australia, and consent to participate in the study. The original aim as stated under Section 2.1.1 was that each respondent would have a minimum of 50 sheep; however, due to the lower number of responses, respondents with <50 sheep were also included. Producers who participated in the survey were offered a 50 AUD grocery shopping voucher as a token of appreciation for their time. 

#### 2.3.2. District Veterinarians

The survey of DVs was undertaken in 2021 at an in-person NSW DV conference held in Broken Hill. Follow up emails containing the link to the survey were then sent to each LLS district to ensure any DVs that were not at the meeting also had the opportunity to participate in the study. 

#### 2.3.3. Data Collection

The data were collected both online and in person, either through the online survey tool REDCap (V11.2.2 © 2021), a University of Sydney hosted secure web-based software platform used to support data retention for research studies [20,21], or via distribution of hard copies of the survey.

### 2.4. Data Management

The data within REDCap were downloaded directly into Excel (V2109, Microsoft^®^) and the data collected on paper-copies of the surveys were manually entered into Excel. The data were then cleaned to check for duplicate responses, abnormal entries, and missing values. 

The sheep producer survey was completed by 47 participants. But two responses were excluded from analysis as one producer was not from NSW and the other was a duplicate response. Hence the information gathered from 45 respondents was utilized. Responses for the DV survey were received from 26 participants, of which 1 was excluded as they were not employed as a DV. Hence the responses of 25 DV participants were included in the analysis. 

The reported percent response for each question was calculated from the number of responses. Not all participants responded to each question and therefore the number of respondents (*n*) is included together with percent response. The response options for rating and scale questions were recategorized based on tertials, e.g., into low-, medium-, and high-concern or ‘most often’, ‘sometimes’, and ‘rarely’. For semi-closed questions that contained many similar answers, variables were differentiated and recoded into respective categories if biologically appropriate. These questions included those that requested ‘other’ information or asked to ‘please specify’ if they chose the option ‘sometimes’. For example, for the question for sheep producers regarding the period in which they separate sheep with disease from their main flock, responses were recoded into categories based on similar time frames or reasonings. This included answers stating, “Until they are well again” or “Until symptoms of health concerns are resolved” that were then recoded into the same category of “until recovered”. Ordinal values for the five-point Likert scale question regarding producer perception on sheep disease management were condensed into three categories for ease of analysis. The options ‘strongly disagree’ and ‘disagree’ were merged to ‘disagree’, and the options ‘agree’ and ‘strongly agree’ were merged to ‘agree’. ‘Neither agree/disagree’ remained the same. If particular responses represented less than 5% of total responses for any one question, the responses that were biologically similar were merged. 

### 2.5. Data Analysis

Frequency tables were generated for the outcomes of all questions. The distribution of both continuous and categorical variables such as the location of the property or ranking of concern of sheep health issues were assessed using bar graphs. The mean, median, and minimum and maximum values of continuous variables were calculated. 

## 3. Results

### 3.1. Response Rate and District Locations

A total of 45 participant responses from sheep producers were included in the analysis. All the LLS districts of NSW except for the North West district are represented by at least one participant. The number of responses from sheep producers within each LLS district is presented in Figure 1.

Twenty-five responses from DV participants were included in the analysis. Seventeen responses were acquired from DVs at the in-person conference in Broken Hill and eight from the subsequent email distribution of the online link to the survey. The DV questionnaire had a response rate of 41.7% (*n* = 25/60) based on the number of DVs employed by the LLS as of March 2022 (Ison, S, personal communication, 22 March 2022). All LLS districts within the state of NSW were represented in the study. The number of responses by DVs and the LLS districts they represent are shown in Figure 1.

### 3.2. Sheep Producer Survey

#### 3.2.1. Sheep Producer Demographics

The demographics of the sheep producers and their enterprise type are presented in Table 1 and Table 2, respectively. The majority of respondents were male (64.4%, *n* = 29) and aged 51 years or older (68.9%, *n* = 31) (Table 1). The question regarding enterprise type required a Yes or No response and the results are presented in Table 1. Enterprise type was not mutually exclusive. From the respondents, most of the farms were a mixed enterprise (sheep and cattle) (*n* = 27) followed by prime lamb production (*n* = 22), wool production (*n* = 19), and non-wool production (e.g., dorper) (*n* = 9). Twenty-two of the enterprises (48.9%) had Merino sheep, and 64.4% (*n* = 29) had other breeds (Table 1). The median flock size was 1750 sheep, and the median farm size was 625 hectares (Ha) (Table 2).

#### 3.2.2. On-Farm Sheep Health Concerns

Over the three years, 2019–2021, that were considered when responding to the survey questions, the most commonly reported sheep health concerns that had been present on the respondents’ farms were internal parasites (82.2%, *n* = 37), lameness (37.8%, *n* = 17), external parasites (35.6%, *n* = 16), and scabby mouth (17.8%, *n* = 8). The health issues with less than 5% responses were abortion, clostridial disease, footrot, toxicity, Ovine Johne’s disease (OJD), and others. 

#### 3.2.3. Sheep Welfare, Mortality, and Their Financial Impact

The impact of common diseases on the welfare of sheep on their property was rated as low by the majority of respondents (68.9%, *n* = 31), followed by moderate impact (28.9%, *n* = 13). However, almost all producers considered the importance of animal welfare for their enterprise as high (95.6%, *n* = 43). Most producers reported internal parasites (53.3%, *n* = 24), followed by lameness (44.4%, *n* = 20), external parasites (40.0%, *n* = 18), OJD (11.1%, *n* = 5), and other diseases (6.7%, *n* = 3) as affecting animal welfare on their farm.

Each farmer reported the mortality rates for the different classes of sheep on their farm. The average of the reported mortality rates from all respondents is presented in Table 3. Lambs had the highest mortality with a median of 5% and ranging from 0–30%, followed by ewes, with median mortality of 3.0% and ranging from 0–10%. The median ram mortality was 1.0%, ranging from 0–25%. Wethers had the lowest mean mortality with a median of 1.0%, ranging from 0–10%.

The most common factor that caused financial concern to producers for the management of common sheep diseases was fencing (70.5%, *n* = 31), followed by veterinary costs (52.3%, *n* = 23), diagnostic costs (50.0%, *n* = 22), labour costs (44.4%, *n* = 20), and vaccination costs (40.0%, *n* = 18). 

#### 3.2.4. Vaccination

Vaccination was used by most producers (91.1%, *n* = 41) to manage flock health. The most frequently used vaccine was 6-in-1 (57.8%, *n* = 26), which includes five clostridial diseases (black disease, black leg, malignant oedema, pulpy kidney, and tetanus) and cheesy gland for caseous lymphadenitis (CLA). This was followed by 5-in-1 vaccine for clostridial diseases (37.8%, *n* = 17), Gudair (26.7%, *n* = 12) for OJD, Scabigard for scabby mouth (11.1%, *n* = 5), Eryvac for erysipelas arthritis (4.4%, *n* = 2), and ‘other’ (6.7%, *n* = 3). Of the producers that used ‘other’ vaccines these included: Campyvax for campylobacter, GlanEry 7-in-1 B12 for erysipelas arthritis, CLA, and five clostridial diseases of tetanus, pulpy kidney (enterotoxaemia), black disease, blackleg, malignant oedema, together with vitamin B12, Barbervax for barber’s pole worm, and Selovin LA, a selenium supplement. Lambs were administered a second booster vaccination (if required according to manufacturer’s instructions) by 79.5% (*n* = 31) of producers, with 20.5% (*n* = 8) not administering a booster. Most producers provided annual booster vaccinations as per the manufacturer’s instructions to ewes (96.8%, *n* = 30), but this was lower for rams (67.7%, *n* = 21) and wethers (25.8%, *n* = 8). 

#### 3.2.5. Management of Internal Parasites

The majority of producers reported no drench resistance against internal parasites in their flock (80.0, *n* = 36). Thirty-eight producers (84.4%) rotated their drenches. Most producers (72.7%, *n* = 32) followed integrated parasite management (IPM) for control of internal parasites. However, just under half of the producers (46.7%, *n* = 21) performed regular faecal egg counts (FEC) to monitor parasite burden. 

#### 3.2.6. Biosecurity Practices

The main biosecurity practices reported by participating sheep producers are presented in Table 4. Most producers had a farm biosecurity plan (86.7%, *n* = 39). When purchasing new stock, 81.0% (*n* = 34) of producers inspected them for disease but 4.8% (*n* = 2) did not and 14.2% (*n* = 6) only did this sometimes. Further, 39.0% (*n* = 16) of producers did not request a sheep health statement when purchasing new stock. The majority of producers quarantined new stock (71.4%, *n* = 30) for periods of 1 to 93 days, with median times of 28 days (Interquartile Range 14–31 days). Overall, 46.3% (*n* = 19) of producers separated sheep with disease from their main flock while 36.6% (*n* = 15) did this ‘sometimes’. Eighteen of the sheep producers responded to the question on the period that they separate sheep with disease from their main flock. The responses varied, with the majority maintaining separation ‘until recovered’ (44.4%, *n* = 8) followed by ‘weeks’ (22.2%, *n* = 4), ‘culled’ (11.1%, *n* = 2), ‘as long as needed’ (5.6%, *n* = 1), ‘varies’ (5.6%, *n* = 1), ‘3 months’ (5.6%, *n* = 1) and ‘one week’ (5.6%, *n* = 1). Seven producers (17.1%) did not separate sick sheep from the main flock. However, thirty-five of the surveyed farmers (83.3%) utilized separate paddocks for isolation/quarantine of new or sick sheep, while seven (16.7%) did not follow this practice. In terms of the perception of sheep producers that quarantining new stock prior to mix-up with the flock and inspecting sheep for disease prior to purchase were important practices, 93.3% (*n* = 42) agreed, 4.5% (*n* = 2) neither agreed or disagreed and 2.2% (*n* = 1) disagreed.

Most producers used sheep health contractors for disease management including footrot inspections, dipping for parasites, vaccination, and drenching (95.3%, *n* = 41), while 4.7% (*n* = 2) used regular farm labour for these tasks.

#### 3.2.7. Accessibility and Use of Services for Information about Sheep Diseases 

Sheep producers typically sort information about the control/prevention of sheep diseases from DVs (66.7%, *n* = 30), private veterinarians (60.0%, *n* = 27), and ‘other’ sources (33.3%, *n* = 15) (Table 5). Of those who chose ‘other’ responses included the internet (42.9%), local rural suppliers (35.7%), other farmers/neighbours (28.6%), consultants (21.4%), LLS (21.4%), and company representatives (14.3%).

### 3.3. District Veterinarian Survey

#### 3.3.1. The Most Common Reasons DVs Were Called out to Properties in Their LLS District

The reasons why DVs were called out to properties were ranked from highest to lowest. The most frequent reason was for disease outbreaks (72.3%. *n* = 16), then sudden death of livestock (57.1%, *n* = 12), reproduction issues (50%, *n* = 11), husbandry advice (44.4%, *n* = 8), other (35%, *n* = 7), vaccination (33.3%, *n* = 6), nutrition advice (33.3%, *n* = 6), parasitology (31.8%, *n* = 7), lamb mortality (30.4%, *n* = 7) (Figure 2). Responses as “other” included footrot/lameness inspections, reduced productivity, animal welfare, plant poisonings, and dog predation.

#### 3.3.2. The Most Common and Difficult Sheep Diseases to Control from the DV Perspective in Their LLS District

Out of five endemic sheep diseases listed on the questionnaire (OJD, internal parasites, external parasites, footrot, and clostridial diseases), DVs were asked on a scale of one to five to describe how common they were in their district, while also considering how difficult they were to control (one being severely concerning and five being of no concern). District veterinarians ranked footrot as the most common and concerning in their district (20%, *n* = 5). This was followed by clostridial diseases (16%, *n* = 4), external parasites (12%, *n* = 3), OJD (8%, *n* = 2), internal parasites (8%, *n* = 2), and other (4%, *n* = 1). The respondent that selected “other” identified this as lameness. 

### 3.4. Common Questions within the Surveys of Sheep Producers and DVs

#### 3.4.1. Areas of Sheep Health of Concern to DVs and Sheep Producers 

When recategorized as tertiles and ranked in accordance with percentage of respondents for each option, the sheep health issues considered to be of the highest concern by DVs within their district were husbandry advice (43.8%, *n* = 7), nutritional advice (42.9%, *n* = 9), lamb mortality (40%, *n* = 8), disease outbreaks (38.1%, *n* = 8), vaccination (35%, *n* = 7), parasitology (33.3%, *n* = 7), reproduction (31.8%, *n* = 7), sudden death of livestock (31.6%, *n* = 6), other (10.5%, *n* = 2), and trauma (10.5%, *n* = 2) (Figure 3A). Responses of “other” included ewe perinatal mortality, accepting stock death in dry times, and poisonous plants.

Sheep farmers were asked to scale the major sheep health problems facing sheep producers in their LLS district from 1 of greatest concern, to 10 of least concern. Based on the percent of responses for the level of concern recategorized to tertiles, parasitology was the highest (50.0%, *n* = 22), followed by nutrition advice (47.7%, *n* = 21), disease outbreaks (45.5%, *n* = 20), husbandry advice (40.5%, *n* = 17), reproduction (37.2%, *n* = 16), trauma (37.2%, *n* = 16), vaccination (36.3%, *n* = 16), lamb mortality (32.6%, *n* = 14), sudden death of livestock (32.6%, *n* =14), and other (31.25%, *n* = 5) (Figure 3B). 

#### 3.4.2. Areas Where Sheep Producers Would Benefit from Increased Knowledge as Identified by the DVs and the Sheep Producers

The majority of DVs identified that sheep producers would benefit from increased knowledge of parasite management (88%, *n* = 22), biosecurity (80%, *n* = 20), quarantine (76%, *n* = 19), common diseases (76%, *n* = 19), reproduction (68%, *n* = 17), correct vaccine use (64%, *n* = 16), lamb mortality (64%, *n* = 16), nutrition (64%, *n* = 16), which vaccine to use (60%, *n* = 15), sudden death of livestock (52%, *n* = 13), parasite life cycles (48%, *n* = 12), sheep husbandry (44%, *n* = 11), trauma (20%, *n* = 5), and other (16%, *n* = 4) (Figure 4). The reasons that DVs explained for selecting “other” were welfare obligations, lameness issues, and footrot/regulations. 

A benefit of increased knowledge of sheep production by sheep producers was identified in the following areas: nutrition (55.6%, *n* = 25), parasite management (53.3%, *n* = 24), common diseases (51.1%, *n* = 23), lamb mortality (48.9%, *n* = 22), reproduction (46.6%, *n* = 21), sudden death of livestock (37.8%, *n* = 17), biosecurity (33.3%, *n* = 15), parasite life cycles (31.1%, *n* = 14), which vaccine to use (28.9%, *n* = 13), correct vaccine use (24.4%, *n* = 11), husbandry (24.4%, *n* = 11), quarantine (15.6%, *n* = 7), trauma (8.9%, *n* = 4), none (4.4%, *n* = 2), and other (2.2%, *n* = 1) (Figure 4). The respondent that selected “other” listed autopsies as an area for increased knowledge. 

#### 3.4.3. Preferred Methods for Dissemination of Information to Sheep Producers, and the Communication Systems Used by DVs

The methods sheep producers preferred to receive information about sheep health were through email (77.8%, *n* = 35), meetings and workshops (60.0%, *n* = 27), in person (51.1%, *n* = 23), pamphlets and written media (44.4%, *n* = 20), industry bodies and groups (33.3%, *n* = 15), government bodies (28.9%, *n* = 13), and via phone call (22.2%, *n* = 10). 

The methods DVs preferred to use to disperse sheep health information to sheep producers were via phone call (56.0%, *n* = 14), in person (40.0%, *n* = 10), through industry body collaboration (4.0%, *n* = 1), and other (4.0%, *n* = 1). One DV selected “other” and ranked it highest, identifying Facebook as their most common method of sharing information.

## 4. Discussion

The main objectives of this study were to gain an insight into the perceptions of NSW sheep producers and DVs about sheep health and the on-farm disease management practices of commercial sheep producers. The frequency and the methods used for contact between sheep farmers and DVs were also explored. To our knowledge this is the first study that has investigated and compared these aspects of sheep health management in NSW. A finding of particular interest was the dichotomy of the farmers’ preferred mechanisms for receiving information and the most common communication methods DVs use to distribute information. By understanding these perceptions and preferences, government agencies, industry bodies, and veterinarians are better able to offer information tailored to the needs of sheep producers that is disseminated through their preferred mechanisms. This study was also an opportunity to compare aspects of sheep production and health that were of concern to the sheep farmers and veterinarians, identifying where differences exist. This has identified the need to provide information that is important to the individual farmer through their preferred medium. This will help increase engagement between the sheep producer and DVs, resulting in improvements in on-farm management of sheep health. 

### 4.1. Areas of Sheep Health Concern

All nominated areas of sheep health within the survey were of considerable concern to over 30% of DVs (except for trauma) and sheep producers. However, the main areas of concern differed between the two groups. For DVs the areas of greatest concern were in providing advice on sheep husbandry, nutrition, and lamb mortality, whereas for sheep producers the most concerning sheep health issues were parasitology, nutrition advice, and disease outbreaks. The shared concern around sheep nutrition highlights the need for greater education and support in this area for both NSW sheep farmers and DVs. Notably the period which this study was focusing on (i.e., 2019–2021) coincided with a series of high rainfall events and severe floods across NSW after many years of drought. Hence, with the potential for more regular drought and flooding events, there is an urgent need for support for the nutritional management of sheep and other livestock [22], including planning for preparedness in both the short and longer term. Further study into nutritional management through extensive dry periods and floods would benefit sheep producers in tackling these problems. This highlights the multi-faceted approach that may be required to improve sheep health outcomes in NSW.

The discrepancy between the greater concern of DVs around animal husbandry and lamb mortality compared to the lower priority of these concerns for the farmers is interesting. It could reflect an overall concern of DVs of perinatal mortality which has been identified as the number one issue of economic impact in the sheep industry [23]. Hence there may be a need for additional education for the NSW sheep farming community on the management and minimization of lamb mortality.

The greater concern of sheep farmers compared to DVs about parasitology and disease outbreaks may indicate a need for extra support or targeted training of veterinarians in these areas that will specifically meet the needs of the farmers in their LLS district. However, DVs also identified that sheep producers would benefit from increased knowledge about parasite management, so it is a common area recognized for deeper education and improved understanding. This may also be an avenue for veterinarians to reiterate the links between biosecurity and animal health, with an aim to minimize disease outbreaks, which was also an identified area of concern for farmers [24]. 

### 4.2. Access to Veterinarians

More than 50% of sheep farmers perceived their local veterinary services to be quite a long distance (far) from them. Further, most producers reported ‘moderate’ as opposed to ‘easy’ access to veterinary assistance. This concurs with previous Australian-based studies which have identified the comparative remoteness of sheep farms from veterinary services [7,25]. In Western Australia the distance to the nearest veterinarian was the predominant predictor of whether sheep and cattle producers would contact a vet in the event of a significant disease outbreak [7]. However, whether the availability and the accessibility of veterinarians were associated with contacting a veterinarian in this NSW based study was not evaluated but is worthy of further investigation. 

### 4.3. Vaccination

Most sheep producers that participated in this study vaccinated their sheep. However, where a secondary and booster vaccination is required (3-in-1, 5-in-1, 6-in-1, and Eryvac), one-fifth of producers did not administer the second vaccine to their lambs and just over 25% did not deliver an annual booster vaccination to wethers. In a similar vein, 17% of Merino lamb producers reported that they provide only a single vaccination for clostridial diseases [13], despite a booster being recommended. This reiterates the need for further education of sheep producers on the importance of following the recommended vaccination program and, the instructions for vaccine use [13]. Future studies could also explore the relationship between vaccine use and, farmer and farm demographics, with an aim to provide more targeted education programs. 

### 4.4. Internal Parasite Control

Internal parasites are a long standing significant and endemic sheep health issue within Australian sheep flocks [6]. It was not surprising then that in this study parasitology was identified as the most common sheep health issue during the study period. As the development of anthelmintic resistance continues to rise, implementing sustainable management strategies against internal parasites is essential [26]. While most producers indicated that they rotate their drenches, only half of them regularly use FEC. As FEC objectively test drench efficacy [27], these producers may not be aware of the effectiveness of their drenches. However, it is important to note that this study had a relatively high proportion of participants from low rainfall regions such as Western NSW (9 of 45 respondents, Figure 1), that typically receive only 290–300 mm rain/annum [28]. In those areas internal parasites may not be of sufficient concern to warrant FEC. 

### 4.5. Biosecurity

Biosecurity practices varied amongst sheep producers. While the majority of producers reported having a biosecurity plan, separating diseased stock from the main flock was undertaken by less than half of respondents, and just over one third of respondents separated stock with disease only ‘sometimes’. This is problematic as mixing sheep with an infectious disease within the main flock can facilitate the rapid spread of the disease-causing pathogens throughout the flock [14]. Disparity between the biosecurity practices producers used compared to their perceived importance of those practices were apparent. Interestingly, just under one half of producers separated diseased stock and 81% inspected new stock for common diseases prior to purchase, yet more than 90% of respondents acknowledged the importance of these biosecurity practices. Such inconsistency between utilizing biosecurity procedures and their perceived importance has also been reported by others [5,29]. For example, beef producers in Australia’s Northern and Southern beef zones were aware of the disease risks when introducing new livestock from saleyards or other farms to their property but they did not consider these risks to be a major concern and perceived them to have little impact on farm productivity [25]. The sheep producers’ perception of biosecurity risks was not investigated in this study, but it could be speculated that they may perceive their livestock to be at low risk of disease incursion as their farms are often remote and Australia is free from many diseases of international concern such as foot and mouth disease [30]. However, further investigation is required to better understand the producers’ perceptions towards the risk of disease occurrence and the influence this may have on their use of biosecurity practices. 

The maintenance of stock proof fences is an important biosecurity practice within sheep farming systems [25]. Stock proof fencing prevents sheep from straying and minimises the introduction of pathogens, for example lice, from nearby properties. In this study more than 70% of producers rated fencing as the costliest aspect of disease management. Hence, NSW sheep producers may benefit from greater support, possibly through local government subsidies, to assist in maintaining their fences. 

### 4.6. Mortality Impact

The average annual lamb mortality rate reported by producers in this study was similar to another recent report of just under 10% lamb mortality on NSW sheep farms [13]. However, earlier studies report higher mortality percentages within Australian sheep flocks [31,32]. For example, 10–30% lamb mortality was observed in a 2011 study of Western Australian sheep producers [32] while a study conducted in 2007 reported 31.5% mortality in twin lambs and 16.5% in singles born from mature ewes [33]. Similarly, reports from the UK indicate typically higher lamb mortality of between 15–20% [34,35]. The reasons for the comparatively lower rate of lamb mortality in this study are unknown but given its significant impact on flock efficiency [31] these findings are worthy of further investigation. 

### 4.7. Methods of Communication

A variety of communication skills are essential for veterinarians to effectively disseminate knowledge while developing a strong relationship with their clients [8,36]. Email was identified as the preferred communication method by more than three quarters of the sheep farmers surveyed, followed by meetings and workshops, while more than half of the producer responses also selected in-person discussions, whereas, receiving advice over the phone or through discussions in town hubs for example at the sale yards and pubs, were the least preferred communication mediums of the sheep producers. In contrast, DVs preferred to communicate with producers via phone call and in person. These were not the preferred communication methods identified by farmers. Further, email communication, which was preferred by producers, was not commonly used by DVs. These discrepancies deem the need for further exploration of the avenues of engagement which may be more tailored to the preferred modality of the client [8]. Particularly, email and communication opportunities for sheep producers through meetings and workshops may achieve improvements in sheep health and may also strengthen the farmer/veterinarian relationship. However, both methods are likely to be more time consuming for DVs in comparison to a phone call, especially as the latter can be accomplished while performing other tasks, such as a hands-free phone call while driving. It should be noted that the impact of COVID-19 on the responses about methods for the dispersion of information within these surveys is unknown. This would be a valuable question to include in future surveys.

### 4.8. Areas Where Further Information Would Benefit Sheep Producers 

All participating DVs and most sheep producers indicated that the latter would benefit from greater knowledge and access to information on sheep health. While this shared acknowledgement is a solid basis for more regular dissemination of information between the two groups, there is some disparity between the areas of sheep health that each group perceived as being the most critical. For DVs the priority areas for improving the knowledge for sheep producers were parasite management, common diseases, and biosecurity. In contrast farmers identified more diverse areas but the top three were nutrition advice, parasite management, and common diseases. As both groups identified benefits for producers in having a greater understanding of parasite control and common diseases, there is opportunity for increased collaboration in these areas. 

Conversely, DVs rated biosecurity as the third area in which they perceived producers would benefit from further information, while advice on sheep nutrition was the third category identified by the farmers. This highlights areas where veterinarians could more frequently provide advice and assistance to NSW sheep farmers. This may involve collaboration with agronomists and nutritionists to deliver the level of information that is needed. Further, extension programs could include additional emphasis on the value of biosecurity through demonstration of the close links between farm biosecurity, sheep health outcomes [37] and lost income. 

### 4.9. Study Limitations 

This study included some limitations and biases that should be considered when interpreting the findings. A key limitation was the small sample size achieved with the survey for sheep producers. While 45 respondents were included, in total 11,710 NSW businesses are involved in sheep and/or lamb production [38]. Given this, results from this study cannot be accurately applied to the entire population of sheep farmers in NSW and interpretations from this study should be considered in this context. The study included producers from all regions of NSW except for the North West LLS district. Therefore, broadly, the findings are representative of farms from most geographical locations within NSW but due to their unique circumstances, extrapolation of these findings by producers in the North West district should be treated with caution. Similar caution should also be used when considering these findings in relation to sheep production in other Australian states and throughout the world. A larger cohort of study participants may have been recruited if the study had been completed over a longer period of time and/or if it had been distributed both on-line and in person as with the DV survey. However, the presence of COVID-19 in NSW during the time of the survey also largely contributed to reduced opportunities for its in-person distribution [39]. 

The study outcomes may contain some bias as the survey was voluntary and was predominantly distributed through NSW sheep societies and by local DVs. This can introduce non-response bias, where respondents may have more interest in the topic compared to non-responders. However, these modes of distribution did capture responses from a wide variety of LLS districts during a state-wide lockdown. The DV survey, which had a higher response rate (41.7%) compared to the producer survey, was distributed in person at the 2021 Broken Hill DV Conference. This may have generated bias by increased participation of DVs from the Western LLS district. To mitigate this, all NSW LLS offices were contacted via email offering DVs online participation. However, as only 8/25 responses (32%) were returned via this method, bias of responses from some regions represented at the DV conference cannot be ruled out.

Finally, self-reporting surveys such as those used in this study are prone to information bias [40]. This can include the impact of the recall period, where recall accuracy tends to decline with time and even within short periods, for example, recalling events from yesterday compared to the previous week [40]. The current study involved recall from the previous three years, a timeframe that is likely to reduce the accuracy of some responses. Selective recall, in that some events are more readily recalled than others and adjustments of responses so they more closely align with social desirability, even when the data are presented anonymously [41], may have also introduced some bias in the reported data. 

## 5. Conclusions

Sheep farmers in NSW most often sort veterinary assistance for disease outbreaks, sudden livestock deaths, and reproduction issues. However, access to veterinary services and to DVs that need to travel to visit sheep farms is seen as an issue and could be improved. While DVs prioritized providing advice on husbandry and nutrition, sheep producers expressed greater concern over parasitology and disease outbreaks. These discrepancies underscored the importance of tailored information and dissemination strategies that would effectively address the specific needs and priorities of both groups. Additionally, vaccination and vaccination schedules, internal parasite control, and biosecurity practices are areas for further attention and education for the improvement of overall flock health and farm productivity. Further, a closer alignment of communication strategies with the preferences of the target audience may assist in maximizing engagement and information uptake.

The insights gained from this study emphasize the multi-faceted nature of sheep health management and the importance of collaborative efforts between sheep farmers, DVs, government agencies, and industry bodies to effectively address existing challenges. By tailoring information dissemination strategies, enhancing access to veterinary services, and prioritizing education in key areas of concern, stakeholders can work towards improving sheep health outcomes and fostering sustainable sheep farming practices across NSW.

## Figures and Tables

**Figure 1 animals-14-01249-f001:**
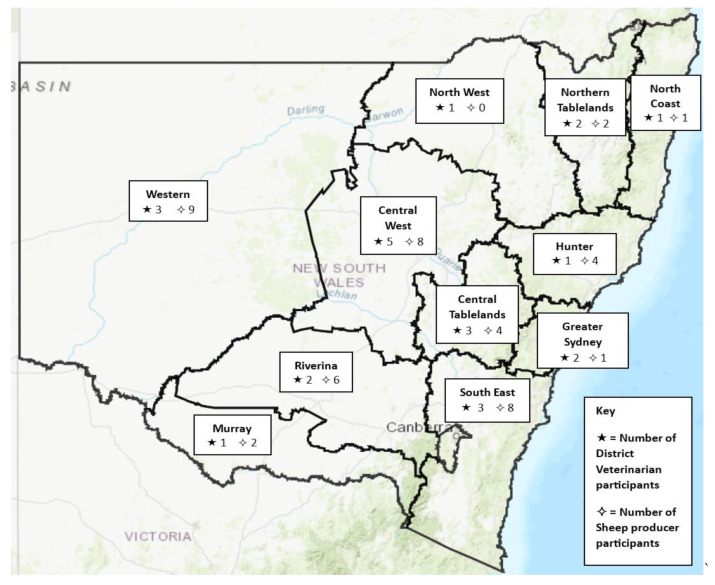
Map of NSW showing number of survey participants of sheep farmers (*n* = 45) and district veterinarians (*n* = 25), across Local Land Services districts *. * One district veterinarian survey participant was excluded as they did not specify their LLS district.

**Figure 2 animals-14-01249-f002:**
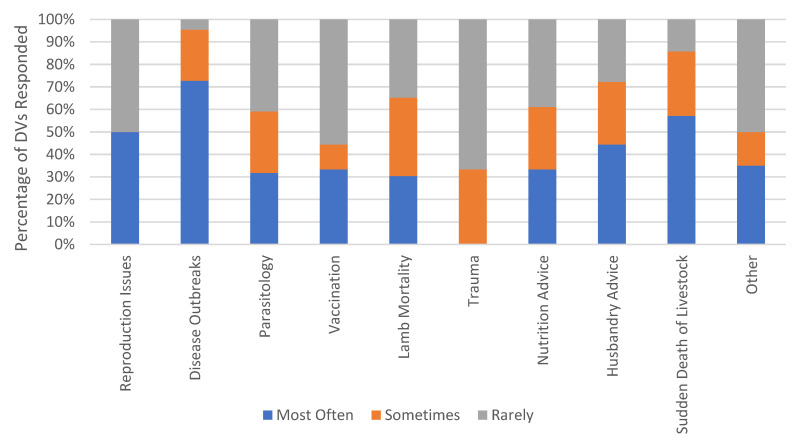
Ranking of the reasons district veterinarians (DVs) were called to properties in their Local Land Services district.

**Figure 3 animals-14-01249-f003:**
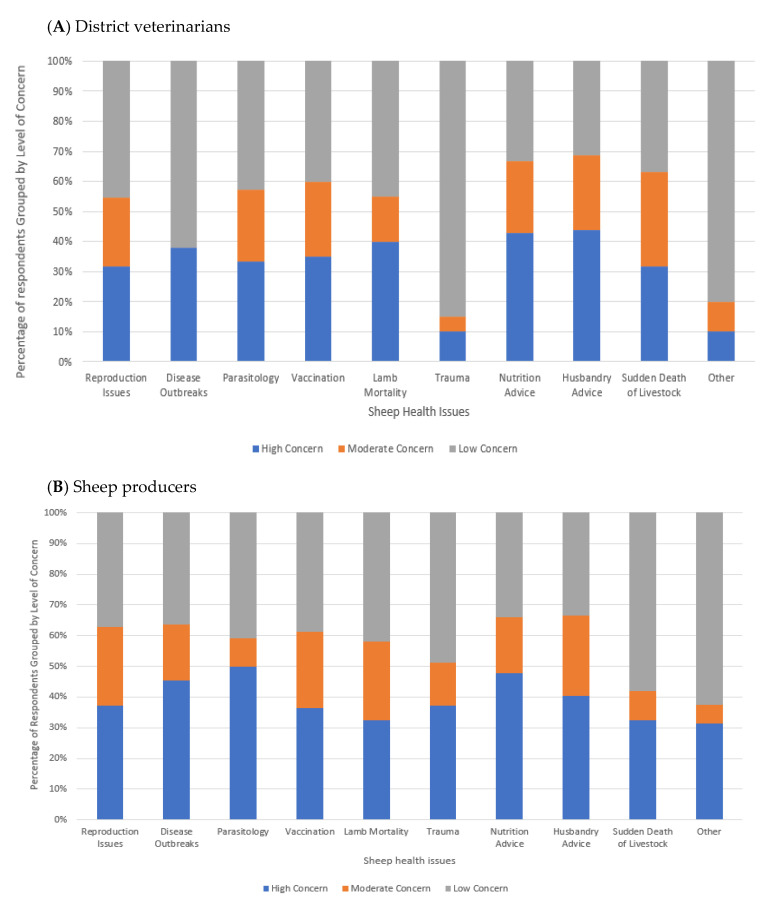
Perceived level of concern for sheep health problems of district veterinarians (**A**) and sheep producers (**B**).

**Figure 4 animals-14-01249-f004:**
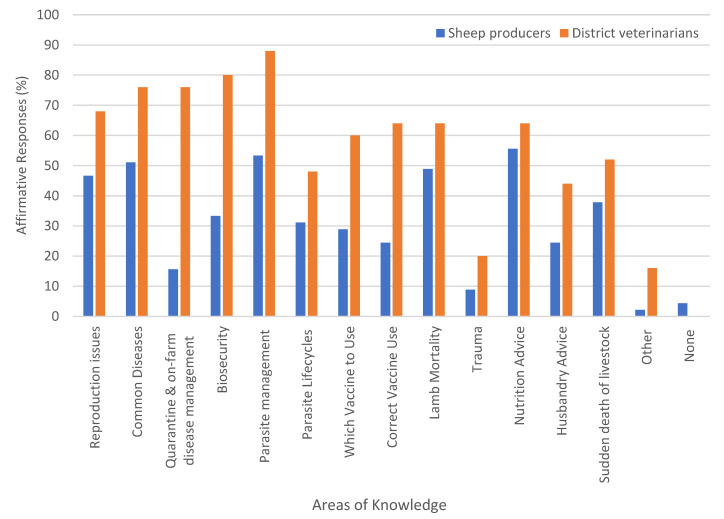
Sheep health issues of which sheep producers would benefit from increased knowledge, as perceived by both the sheep producers and district veterinarians.

**Table 1 animals-14-01249-t001:** Producer and enterprise demographics of sheep farming operations in NSW.

Variable	Category	Frequency (*n*)	Percent (%)
Sheep producer gender			
	Female	16	35.6
	Male	29	64.4
Sheep producer age			
	≤30 years	5	11.1
	31–50 years	9	20.0
	≥51 years	31	68.9
Enterprise type #			
Prime lamb production			
	Yes	22	48.9
	No	23	51.1
Cropping (other than fodder)			
	Yes	7	15.6
	No	38	84.4
Mixed (sheep and cattle)			
	Yes	27	60.0
	No	18	40.0
Wool production			
	Yes	19	42.2
	No	26	57.8
Non-wool (e.g., dorper)			
	Yes	9	20.0
	No	36	80.0
Sheep breed			
Merino	Yes	22	48.9
	No	23	51.1
Other breeds			
	Yes	29	64.4
	No	16	35.6
Type of flock	Self-replacing	32	74.4
	Purchasing	11	25.6

*n*: Number of farms; #: Enterprise types are not mutually exclusive.

**Table 2 animals-14-01249-t002:** Descriptive analysis for continuous variables of farm demographics.

Variable	Mean	SD	Min	Q1	Median	Q3	Max	*n*
Property area (ha)	4674.	9341	5.0	135	625	4328	44,300	44
Total number of sheep on the property	2695	3597	4.0	191	1750	4115	18,480	45
Property altitude (m)	427	244	40	246	350	600	1028	42
Annual average rainfall (mm)	556	247	12	317	556	705	1050	33

SD: Standard deviation; Min: Minimum value; Q1: First quartile; Q3: Third quartile; Max: Maximum value; *n*: Number of farms.

**Table 3 animals-14-01249-t003:** Descriptive analysis for continuous variables of average yearly percent mortality as reported by participating NSW sheep farmers.

Variable	Mean	SD	Min	Q1	Median	Q3	Max	*n*
Lamb mortality	7.4	7.2	0.0	1.3	5.0	10.0	30.0	40
Ewe mortality	3.2	2.3	0.0	2.0	3.0	5.0	10.0	39
Wether mortality	1.7	2.7	0.0	0.0	1.0	2.0	10.0	28
Ram mortality	3.0	4.9	0.0	0.0	1.0	5.0	25.0	32

SD: Standard deviation; Min: Minimum value; Q1: First quartile; Q3: Third quartile; Max: Maximum value; *n*: Number of farms.

**Table 4 animals-14-01249-t004:** Biosecurity measures used by sheep producers in NSW.

Variable	Categories	Frequency (*n*)	Percent (%)
Farm biosecurity plan	Yes	39	86.7
	No	6	13.3
Inspect new stock for common diseases prior to purchase	Yes	34	81.0
	No	2	4.8
	Sometimes	6	14.2
Request a Sheep Health Statement (SHS) prior to purchasing new stock	Yes	25	61.0
	No	16	39.0
Quarantine new stock	Yes	30	71.4
	No	5	11.9
	Sometimes	7	16.7
Separate sheep with disease from the main flock	Yes	19	46.3
	No	7	17.1
	Sometimes	15	36.6
Use quarantine areas/paddocks for new stock or sick animals on the farm	Yes	35	83.3
	No	7	16.7

**Table 5 animals-14-01249-t005:** Accessibility of services for sheep health management by sheep producers in NSW.

Variable	Categories *	Frequency (*n*)	Percent (%)
Obtained information for the control/prevention of disease from a district veterinarian	Yes	30	66.7
	No	15	33.3
Obtained information for the control/prevention of disease from private veterinarian	Yes	27	60.0
	No	18	40.0
Distance to veterinary services	Near	19	41.9
	Far	24	58.1
Are there adequate veterinary services in your district to manage disease occurrence and outbreak?	Yes	32	71.1
	No	13	28.9
How would you describe the ease of accessing veterinary assistance?	Easy	20	45.5
	Moderate	24	54.5
What is the availability of education programs/services for sheep management and disease control within your district?	Adequate	25	55.6
	Lacking	9	20.0
	Could be improved	11	24.4
When was the last time you contacted a vet for assistance/advice regarding the health of your sheep?	<3 months ago	14	31.1
	6 months ago	10	22.2
	1 year ago	10	22.2
	>2 years ago	7	15.6
	Never	4	8.9

* Only the categories that were selected by respondents are presented.

## Data Availability

The original contributions presented in the study are included in the article, further inquiries can be directed to the corresponding author.

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
