# Peer review of "Perceptions of Sheep Farmers and District Veterinarians towards Sheep Disease Management in New South Wales, Australia"

_animals, 2024, doi:10.3390/ani14081249_

Round 1

Reviewer 1 Report

Comments and Suggestions for Authors

This paper is generally well-written and easy to read and understand.

This study was conducted during Covid time, and limitations have been explained, thus the low sample size is understood.

Though the study is focused on a specific condition, the information gained can be utilised by other institutions facing similar issues when dealing with production farmers.

Some comments and questions are as attached in the document.

Reviewer 2 Report

Comments and Suggestions for Authors

Overall comments: This is a descriptive paper on sheep farmers’ and veterinarians’ perspectives of sheep health management. The results are clear, and I only have some minor comments. My main comment is about the grammar / punctuation used, as this impacts the readability of the paper.

The paper should be read through, and the grammar/punctuation amended. There are many cases where commas or other punctuation may have not been used appropriately. Some of these areas include but are not limited to lines 28, 33, 36, 47, 49, 60, 63, 67, 72, 94, 100, 144, 504.

Introduction:

L46 - The first sentence could have a reference for the information you provide.

L79 – perceived by who?

L100 – Perhaps either say why this information from the equine industry is relevant to your work or remove this.

L109 – maybe there could be better wording than “tease out” here – something like “understand”?

Methods:

L103 – what is a semi-closed question?

L104 – a dichotomous question is a closed question – so this may be tautological and unnecessary to state.

L104 – is a copy of the questionnaire included for readers to view? That would be useful to evaluate the paper further.

L108 – Consider the grammar here: surveys were manually entered… Data were…

L182 – Consider the wording of this sentence: response to each question was calculated…

Results:

210 – Consider the wording of this sentence: Northwest districts were represented…

217 – districts is spelt wrong

Discussion:

L574 – producers’

L588 – could you also discuss mortality in relation to other countries?

L598 – could Covid have impacted the respondents’ preferences for the communication method?

L665 – other limitations to consider could be self-report and social desirability bias.

Comments on the Quality of English Language

The quality of the English language is good, but the grammar could be improved to aid readability (see above comments).

Reviewer 3 Report

Comments and Suggestions for Authors

Dear Authors,

Thank you for your study. It was very interesting to read about the management practices on sheep farm in SW Australia, and about the perception of farmers and veterinarians. I think the work you did was very important and interesting. Generally, I would recommend to detail the surveys better so it is easier to understand the results you present (see my specific comments below).

Abstract

Some results are not consistent with text, please review (e.g., L35: 65.7%, while 70% is reported in the manuscript)

Introduction

It seems you used dairy cattle references for sheep (e.g., [2] Denis-Robichaud et al. on L50-53,76-77). I would change the wording so it is clearer you refer to research on dairy cattle in these cases.

L64-71: There are a lot of repetitions in this paragraph; consider rephrasing to avoid this.

L101: remove comma after ‘.’

L110-115: The way the framework is explained seems more like how you developed the surveys: this would fit better in the Methodology section, but it could also be rephrased to be part of the background.

Objectives

L116-117: I  would suggest ‘Describe’ as ‘Understand’ is not a very specific objective. Additionally, what do you mean by ‘the impact of disease at the farm level’? What disease? Is it also linked to the producers’ practices and perceptions? I think this sentence needs to be rephrased for clarity.

L118-19: Again, ‘Describe’ seems more appropriate.

Materials and Methods

L131-133 and L 143-147: The inclusion criteria would fit better in the survey distribution section.

For the questionnaire design, it would be interesting to have more details of what was asked (and how):

-          L133-134 and L147-148: Add the number of questions for each question type.

-          Could you make the surveys accessible?

-          I think it would be important to detail how questions were asked (both producers and DVs questionnaires). When I read the results, I wonder what was asked to support your findings (see results section)

Section 2.3 Survey distribution can be part of ‘data collection’ (section 2.2).

L181: What do you mean by anomalies?

L184-186: As the questions have not been detailed, it is unclear what this means (and how many questions have been recategorized). This could be described better.

Results

L200-204: This could be removed.

L209: Could you estimate a response rate here too?

L231-233: The enterprise type is not clear: I guess they are not exclusive, but this should be made clearer in how the sentence is written (also in Table 1 as it must be stand-alone).

L234: ‘Forty-nine’

L234-235: The breed information is confusing in Table 1 and it is not clear how you got to the % presented in the text.

L239-244: It is not cleared what was measured here an how. A better description in the Methods (What was the question? How was is analyzed?) Did you measure concerns or frequency of disease? The title of the sub-section should represent what was measured.

L249-250: Again, how was this measured? I would be surprised a farmer would say he/she consider the welfare of their animals as unimportant. This would need more details in question/analyses, and perhaps in he way this result is presented.

L250-253: How did you measure this?

L254-258: How was this measured? Did you have access to farm records? If this was reported by farmers, I would recommend changing ‘estimated’ (which sounds like you measured it) by ‘reported by farmers’.

Table 1: You could consider adding a column for missing values: it gives information on the completeness of the data and is easier for the reader than adding numbers throughout.

Table 2:

-          I would also recommend to remove decimals (they make the table very busy and don’t add much information to the reader).

-          I thought you had an inclusion criteria of 50 sheep, but a farm with 12: is it a mistake?

-          In the legend, you report N: it should be ‘n’ (throughout the manuscript) as it is unlikely the whole sheep farm population, but this column is not in the table. It would be nice to have it (or a missing value column).

Table 3:

-          If these are reported by farmers, mention in the title (not estimated, but reported by farmers)

-          Replace Farms for n (or keep uniform in Table 2). Your legend still has ‘N’

L328-333: I am guessing farmers use more than one vaccine (as the total is > 45). Would it be pertinent to describe the combinations? Were these vaccines used for all animals?

L352-354: Here and throughout the manuscript:

-          Please report decimals the same what (1 or none, but always the same).

-          I did not double check everywhere, but it is important you verify the sums (n and % add up). Here, the n = 42 (I expect there were 3 missing values), but sum of % is 100.3%...

L355: Sheep health statement when they purchase new animals?

L375: I am not sure if this is the end of the sentence or if there is text missing. It would also be nice to know how many farmers mentioned each resource.

Table 4: Yes/No/Sometimes are not exclusive categories. Adding information on the questions and analyses would help here, but the categories All the time/Sometimes/Never are likely more appropriate.

Table 5: The categories for contact with a veterinarian have big gaps: is it how it was asked?

L397-402: The way the question was worded and analyzed would help understanding this: were they asked to rank them? From Figure 2, it seems it was a list of conditions and they had to give a frequency, but the way the results are phrased, it seems they were asked what are the common reasons you were called out to a sheep farm.

L408-412: what do you mean by difficult in the title? Is it the same as ‘severe’ in the paragraph?

Figure 3: I am not sure ranking is the appropriate term, but again, this could be clarified in the Methods.

L426-431: The wording here would benefit some change. Looking at the Figure, it seems you mean ‘The proportion of sheep farmers who considered the following aspects of sheep health to be of high concern for sheep enterprise’s in their LLS district:’ (did they rank them or did they categorize their concern for each condition?)

Discussion

L480: What interaction do you mean here?

L482-484: I would have liked to read more about the implications of this findings, and hypotheses about why farmers and DVs have such different preferences. Also, what are the likely impacts of this discrepancy? You mentioned DVs could prepare email material to disseminate, but would this mean it is free? It would be less likely to be tailored to each farm, and DVs would not be able to assess if the information was well understood by farmers. It definitely raises a need for better communication though.

L497-500: I would suggest to remove results from the discussion section. This should focus on interpretation and implications of findings. I would also argue that even though there were differences, if you added moderate concern, the patterns were similar.

L538-546 (and throughout the discussion): Please refrain to present results in the discussion.

L589: Consider replacing ‘experienced’ by ‘reported’

L588: This section is highly speculative. Unless you had access to farm reports, you should use the cited literature to highlight the potential for memory bias in your data.

L628: Are veterinarians the stakeholders supporting sheep nutrition in Australia? The importance of nutritionists/ agronomist, and their collaboration with veterinarians could be highlighted here.

L640+: The comparability of the ranking questions between the 2 surveys could be discussed too. We don’t have the information now, but it is possible that the questions have influenced how the results from the two groups compare. Just something to consider.  

L649: To other regions and worldwide too.

L652: Why would have in person questionnaire been better attended?

L664-665: This should be in the results, not the discussion.

Conclusions

L672-673: The distance did not seem such a problem for respondents (only easy and moderate access were selected by respondents).

Reviewer 4 Report

Comments and Suggestions for Authors

The study by Boyd-Weetman et al. surveyed sheep farmers and veterinarians in NSW, Australia, aiming to identify prevailing health management practices on sheep farms. The study uniquely explored differences in perceptions between farmers and veterinarians regarding various aspects of sheep farming practices. While the study focuses on a specific geographic area, its findings could inspire other researchers to apply similar strategies in their respective regions. The manuscript was well written, engaging, and easy to follow. Below are some comments and suggestions for your consideration.

The introduction is well-structured and effectively establishes the rationale for the study with clearly defined objectives.

Line 52: Please use the plural form of the verb with "data." This correction should be applied consistently wherever applicable in the manuscript (lines 179,180, and others).

The data management and analysis were sufficiently described.

Line 207–209, 213–214: The information regarding the exclusion of survey participants may be relocated to the data management section.

Figure 1 effectively illustrates the distribution of participants based on location.

Line 234: Fory?

Consider using the past tense when describing the results.

Line 506: multi-facilitated approach?

The discussion was adequately referenced to justify the results, and the manuscript also acknowledges the limitations of the study.

While the conclusion effectively summarizes the results, it could be more assertive in clearly outlining the key findings of the study, avoiding a restatement of the study's objectives or focus.
